# Longer Photoperiods with the Same Daily Light Integral Increase Daily Electron Transport through Photosystem II in Lettuce

**DOI:** 10.3390/plants9091172

**Published:** 2020-09-10

**Authors:** Claudia Elkins, Marc W. van Iersel

**Affiliations:** Department of Horticulture, University of Georgia, Athens, GA 30602, USA; claudia.elkins@uga.edu

**Keywords:** photochemistry, daily photochemical integral, electron transport, quantum yield of photosystem II, chlorophyll fluorescence, photoperiod

## Abstract

Controlled environment crop production recommendations often use the daily light integral (DLI) to quantify the light requirements of specific crops. Sole-source electric lighting, used in plant factories, and supplemental electric lighting, used in greenhouses, may be required to attain a specific DLI. Electric lighting is wasteful if not provided in a way that promotes efficient photochemistry. The quantum yield of photosystem II (Φ_PSII_), the fraction of absorbed light used for photochemistry, decreases with increasing photosynthetic photon flux density (PPFD). Thus, we hypothesized that the daily photochemical integral (DPI), the total electron transport through photosystem II (PSII) integrated over 24 h, would increase if the same DLI was provided at a lower PPFD over a longer photoperiod. To test this, Φ_PSII_ and the electron transport rate (ETR) of lettuce (*Lactuca sativa* ‘Green Towers’) were measured in a growth chamber at DLIs of 15 and 20 mol m^−2^ d^−1^ over photoperiods ranging from 7 to 22 h. This resulted in PPFDs of 189 to 794 μmol m^−2^ s^−1^. The Φ_PSII_ decreased from 0.67 to 0.28 and ETR increased from 55 to 99 μmol m^−2^ s^−1^ as PPFD increased from 189 to 794 μmol m^−2^ s^−1^. The DPI increased linearly as the photoperiod increased, but the magnitude of this response depended on DLI. With a 7-h photoperiod, the DPI was ≈2.7 mol m^−2^ d^−1^, regardless of DLI. However, with a 22-h photoperiod, the DPI was 4.54 mol m^−2^ d^−1^ with a DLI of 15 mol m^−2^ d^−1^ and 5.78 mol m^−2^ d^−1^ with a DLI of 20 mol m^−2^ d^−1^. Our hypothesis that DPI can be increased by providing the same DLI over longer photoperiods was confirmed.

## 1. Introduction

The controlled environment agriculture industry, including indoor plant factories and greenhouses, in the United States (U.S.) spends $600 million annually on the electricity required for horticultural lighting [1]. To reduce these energy costs, it is important to understand how efficiently plants use the light they receive. Only then can strategies be developed to provide lighting when needed, with the intensity needed, to assure that the light can effectively enhance plant growth.

Light (photons) is necessary for photosynthesis; however, light must be provided at appropriate intensities. Too much light may damage plants [2] and too little light will inhibit growth [3]. To understand how efficiently plants use the light they receive, it is critical to study the process of photosynthesis and the light reactions in particular. The energy from photons absorbed by photosynthetic pigments in plants has one of three fates: (1) drive the light reactions of photosynthesis (photochemistry), (2) dissipated as heat (non-photochemical quenching of chlorophyll fluorescence), or (3) re-emitted as light from chlorophyll *a* (chlorophyll fluorescence). These three fates compete, thus, an increase in one process must be accompanied by a decrease in one or both of the other processes [4,5].

Measuring chlorophyll fluorescence is widely used to study the light reactions of photosynthesis. It provides a non-invasive method to quantify the quantum yield of photosystem II (Φ_PSII_), a unitless measure of the efficiency with which absorbed photons are used to drive photochemistry [6]. It can be used to quantify photochemical responses to photosynthetic photon flux density (PPFD). As the PPFD increases, a larger portion of the photosystem II (PSII) reaction centers become closed (unable to accept additional excitation energy). This occurs because once the electron acceptor Q_A_ in PSII has accepted an electron, it cannot accept another until the first one is transferred to the electron carrier Q_B_. The decrease in “open” reaction centers with increasing PPFD results in a decrease in Φ_PSII_ [5]. To safely dissipate much of the excess light energy, plants upregulate heat dissipation, which further lowers Φ_PSII_ [4,5].

Under natural lighting conditions in a greenhouse, Φ_PSII_ of ‘Green Towers’ lettuce decreased asymptotically from 0.81 to ≈0.22 as the PPFD increased from 0 to 1500 µmol m^−2^ s^−1^ [7], while in a separate study Φ_PSII_ of ‘Little Gem’ lettuce decreased to only ≈0.35 at a PPFD of 1500 µmol m^−2^ s^−1^ [8]. For lettuce ‘Tiberius,’ grown under a PPFD of 200 µmol m^−2^ s^−1^, Φ_PSII_ decreased to ≈0.22 at a PPFD of 1400 µmol m^−2^ s^−1^ [9]. This is consistent with our findings that there are substantial differences in the response of Φ_PSII_ and electron transport rate (ETR) to PPFD among lettuce cultivars (unpublished). In addition, leaves acclimate to the light environment they are exposed to: leaves acclimated to high light levels have higher Φ_PSII_ and ETR than leaves acclimated to lower light levels, when measured at the same PPFD [10].

The Φ_PSII_, combined with a known PPFD, can be used to calculate the electron transport rate (ETR) through PSII as Φ_PSII_ × PPFD_absorbed_ × 0.5 (two photons are needed to move one electron through the entire electron transport chain) [4,6,11]. Oftentimes, PPFD_absorbed_ is estimated as 0.84 × PPFD, assuming leaves absorb 84% of the photons that strike the leaf surface, which is a common leaf absorptance coefficient for C_3_ plants [12].

Electron transport rate, in turn, can be used to determine the daily photochemical integral (DPI), defined as the ETR integrated over a 24-h period. Under greenhouse conditions, without supplemental light, the DPI increased asymptotically with increasing DLI. Because those data were collected under natural sun light conditions, potential interactions between photoperiod, PPFD, and DLI could not be determined [7]. Our goal was to quantify the effect of PPFD and photoperiod on DPI, while maintaining a static DLI. Because Φ_PSII_ increases with lower PPFDs, we hypothesized that the DPI would be greater when the same DLI was provided over longer photoperiods at lower PPFDs. To better understand the physiological basis of treatment effects on Φ_PSII_, we also measured Φ_NPQ_ (the quantum yield of non-photochemical energy dissipation in response to light exposure) and Φ_NO_ (the quantum yield of other, non-light induced energy dissipation processes). Combined, Φ_PSII_, Φ_NPQ_, and Φ_NO_ account for the fate of all photons absorbed by light-harvesting antennae surrounding PSII. In addition, we quantified the upregulation of non-photochemical quenching, as compared to the leaf in its dark-adapted state (non-photochemical quenching (NPQ), which is linearly related to heat dissipation) [4]. A better understanding of the relationship between Φ_PSII_, PPFD, photoperiod, DLI, and DPI can lead to the development of more efficient lighting strategies.

## 2. Results

### 2.1. Time Course of Φ_PSII_, Φ_NO_, and Φ_NPQ_

Photochemical responses of ‘Green Towers’ lettuce were measured at two DLIs (15 and 20 mol m^−2^ d^−1^), each applied across photoperiods of 7, 10, 13, 16, 19, and 22 h. The PPFD was constant throughout the entire photoperiod and ranged from 189 to 794 µmol m^−2^ s^−1^, depending on the photoperiod and DLI combination (Table 1).

Figure 1 shows Φ_PSII_, Φ_NO_, and Φ_NPQ_ data of lettuce for three different photoperiods tested at a DLI of 20 mol m^−2^ d^−1^. The Φ_PSII_ gradually increased, and Φ_NO_ and Φ_NPQ_ decreased, during the first hour into the photoperiod. All three quantum yields changed little during the remainder of the photoperiod. With a 22-h photoperiod (and PPFD of 252 µmol m^−2^ s^−1^), Φ_PSII_ was much higher than with shorter photoperiods and higher PPFD. Differences in Φ_PSII_ among lighting treatments were largely due to increased energy dissipation (Φ_NPQ_) in response to the higher PPFDs associated with shorter photoperiods; Φ_NO_ was slightly lower with a 7-h photoperiod (and PPFD of 794 µmol m^−2^ s^−1^), but treatment differences in Φ_NO_ were small compared to those in Φ_NPQ_.

### 2.2. Time Course of the Quantum Yield of NPQ and ETR

The NPQ parameter is a measure of upregulation of energy dissipation relative to the dark-adapted state. Since all plants in this study were exposed to similar environmental conditions before measurements were taken, NPQ data can be directly compared among different lighting treatments [4]. Consistent with treatment effects on Φ_NPQ_, NPQ was about five times higher with a 7-h photoperiod (and PPFD of 794 µmol m^−2^ s^−1^) than with a 22-h photoperiod (and a PPFD of 252 µmol m^−2^ s^−1^). With 13- and 22-h photoperiods, NPQ stabilized within an hour, while NPQ kept increasing slowly throughout the 7-h photoperiod (Figure 2).

Instantaneous electron transport rates stabilized after about one hour and were higher with shorter photoperiods and higher PPFD (Figure 2 and Figure 3). The area under each ETR curve represents the DPI, which decreased with shorter photoperiods.

### 2.3. Φ_PSII_ and ETR as a Function of PPFD

Independent of DLI, the Φ_PSII_ decreased linearly (*r*^2^ = 0.77, *p* < 0.001) from 0.67 to 0.29 as PPFD increased from 189 to 794 µmol m^−2^ s^−1^ (Figure 3A). The trend was the same for plants exposed to either DLI, 15 or 20 mol m^−2^ d^−1^. The Φ_PSII_ decreased by 0.663 per mmol m^−2^ s^−1^ increase in PPFD.

Electron transport rate increased in an asymptotic manner from 55 µmol m^−2^ s^−1^ at a PPFD of 189 µmol m^−2^ s^−1^ to 99 µmol m^−2^ s^−1^ as PPFD increased to 794 µmol m^−2^ s^−1^ (*r*^2^ = 0.58, *p* < 0.001) (Figure 3B). The asymptote of the ETR was 102 µmol m^−2^ s^−1^. As was the case for Φ_PSII,_ the trend was the same for both DLIs.

### 2.4. Daily Photochemical Integral

The DPI increased linearly with longer photoperiods, but this increase depended on the DLI. As the photoperiod increased from 7 to 22 h, DPI increased from 2.6 to 4.5 mol m^−2^ d^−1^ (74% increase) and from 2.8 to 5.8 mol m^−2^ d^−1^ (109% increase) for DLIs of 15 and 20 mol m^−2^ d^−1^, respectively (*r*^2^ = 079, *p* < 0.001; Figure 4). With a 7-h photoperiod, the DPI under the two DLIs was similar, but as the photoperiod got longer, differences in DPI among the two DLIs increased. Under a 22-h photoperiod, a DLI of 20 mol m^−2^ d^−1^ resulted in a 27% greater DPI than a DLI of 15 mol m^−2^ d^−1^. The effect of photoperiod on DPI was greater for a DLI of 20 mol m^−2^ d^−1^ because of the wider range of measurement PPFDs (253–794 µmol m^−2^ s^−1^) and thus Φ_PSII_ values (0.28–0.64), compared to the PPFDs (189–595 µmol m^−2^ s^−1^) and Φ_PSII_ values (0.39–0.68) for a DLI of 15 mol m^−2^ d^−1^ (Figure 3).

## 3. Discussion

### 3.1. Time Course of Φ_PSII_ and ETR

It took approximately 1 h for Φ_PSII_ and ETR to stabilize regardless of photoperiod and PPFD (Figure 1 and Figure 2). Plants typically reach steady-state photosynthesis after 5 to 10 min in the light [13]. By comparison, our data suggest it takes longer for Φ_PSII_ and ETR to reach a steady state. The longer time required for photochemistry to stabilize is likely due to complex regulation of the light reactions, which involves multiple processes at different time scales. First, when dark-adapted plants with fully open reaction centers are exposed to light, there is a rapid (≈1 s) increase in chlorophyll fluorescence (the Kautsky effect). This increase in fluorescence, and decrease in Φ_PSII_, occurs because of a reduction of the electron acceptor Q_A_, resulting in temporary closure of some of the PSII reaction centers [4]. In response to continued light exposure, plants upregulate heat dissipation [14]. This results in non-photochemical quenching of chlorophyll fluorescence and less excitation energy being directed towards the PSII reaction centers, with plants typically reaching a steady-state within 15 to 20 min, although this is species-dependent [4].

Photochemistry produces adenosine triphosphate (ATP) and reduced compounds (ferredoxin, nicotinamide adenine dinucleotide phosphate—NADPH) that are used in multiple metabolic processes. Because levels of ATP and NADPH are relatively stable in light-exposed leaves, photochemistry needs to be in balance with the usage of ATP, ferredoxin, and NADPH [15]. Photochemistry not only supports carbon fixation in the Calvin-Benson-Bassham (CBB) cycle, but also processes such as photorespiration [16], nitrate [17] and sulfate reduction [18], and the Mehler reaction [19]. Steady-state electron transport cannot be reached until all these processes have reached steady state, which requires time. For example, regulation of nitrate reduction depends on both CBB cycle activity and photochemistry. Carbohydrate accumulation, resulting from CBB cycle activity, induces upregulation of nitrate reductase mRNA transcript levels [20], presumably followed by increased levels of nitrate reductase. In turn, nitrate reductase activity is regulated by NADPH [20]. Thus, both the amount and activity of nitrate reductase are directly or indirectly dependent on photochemistry. The interplay among the various processes that depend on the light reactions may explain the relatively slow stabilization of photochemistry.

Differences in Φ_PSII_ were largely due to upregulation of energy dissipation in response to higher PPFD, as indicated by differences in Φ_NPQ_ and NPQ among treatments; differences in Φ_NO_ among treatments were small (Figure 1 and Figure 2). Similar effects of short-term changes in PPFD have been reported for tobacco: Φ_NO_ was ≈0.25 and changed little, while Φ_NPQ_ increased from 0 to ≈0.6 in response to PPFD increasing from 0 to 900 µmol m^−2^ s^−1^ [21]. This raises the question whether it may be possible to reduce the increase in Φ_NPQ_ (and thus NPQ) in response to increasing PPFD (see Section 3.5).

### 3.2. Φ _PSII_ and ETR as a Function of PPFD

The Φ_PSII_ decreased linearly as PPFD increased (Figure 3), which is consistent with previous studies. Weaver and van Iersel [7] reported that the Φ_PSII_ of ‘Green Towers’ lettuce grown and measured in a greenhouse under natural light decreased exponentially as PPFD increased from 0 to ≈1500 µmol m^−2^ s^−1^. These plants were exposed to a wider range of PPFDs than those in the current study, which may explain the differences in trends, linear versus exponential, between the two studies. When comparing their data across the range of PPFDs used in this study (189 to 794 µmol m^−2^ s^−1^), they too saw a roughly linear decline in Φ_PSII_ from ≈0.6 to ≈0.3; further declines in Φ_PSII_ at higher PPFDs were small, with a Φ_PSII_ of ≈0.22 at a PPFD of 1500 µmol m^−2^ s^−1^ [7]. Zhen and van Iersel [10] found that the Φ_PSII_ of sweet potato (*Ipomea batatas*), lettuce, and pothos (*Epipremnum aureum*), grown in a greenhouse but measured in a growth chamber under precisely-controlled PPFDs, decreased exponentially with increasing PPFD, and the rate of decrease was species-dependent. In addition to differences among species, the greenhouse light levels strongly impacted Φ_PSII_; plants that were grown under shade showed a more rapid decrease in Φ_PSII_ with increasing PPFD. This indicates that the photochemistry of plants can acclimate to different lighting conditions.

Two processes contribute to the decreasing Φ_PSII_ with increasing PPFD. First, in response to increased light, a larger fraction of PSII reaction centers will be in a closed state and unable to accept additional excitation energy [4,5]. Second, in response to increased light, plants upregulate photoprotective processes, which result in the dissipation of absorbed light energy in the form of heat. Although the exact mechanisms of heat dissipation are still under debate, they depend on a pH gradient across the thylakoid membrane and may involve the xanthophyll cycle and PsbS protein [14,22,23,24].

The asymptotic increase in ETR with increasing PPFD (Figure 3) also is consistent with previous findings. Weaver and van Iersel [7] likewise found that ETR of ‘Green Towers’ lettuce responded to PPFD with an exponential rise to a maximum but reported a higher asymptote (121 µmol m^−2^ s^−1^). Zhen and van Iersel [10] found that the asymptote of the ETR curve depended on the light intensity plants received prior to the measurements, with plants grown in shade having lower ETR asymptotes than those grown in full sun. Different growing conditions and the ability of plants to acclimate to different light conditions may explain the difference in asymptotes between the current study and that reported by Weaver and van Iersel [7]. Plants in this study were grown in a growth chamber with constant PPFD of ≈250 μmol m^−2^ s^−1^ and a DLI of 12.6 mol m^−2^ d^−1^, while Weaver and van Iersel [7] grew plants in a greenhouse with an average DLI of 13.9 mol m^−2^ d^−1^ and much higher peak PPFDs. It is plausible that those greenhouse-grown plants had a greater photochemical capacity. It is important to note that all plants in our study were grown under the same lighting conditions, then transferred, for one day, to another set of lighting conditions for measurement. Therefore, plants in this study were not able to acclimate to the measurement lighting conditions.

### 3.3. Daily Photochemical Integral

The increase in DPI with longer photoperiods, with lower PPFDs and equal DLI, was expected, because of the increase in Φ_PSII_ with lower PPFD. Since DPI is a relatively new concept, first introduced by Weaver and van Iersel [7], there have not been prior reports of photoperiod effects on DPI. Under sunlight only, with plants exposed to natural diurnal fluctuations in PPFD and a photoperiod of ≈12 h, the DPI at a DLI of 15 mol m^−2^ d^−1^ was 2.75 mol m^−2^ d^−1^ and at a DLI of 20 mol m^−2^ d^−1^ was 3.0 mol m^−2^ d^−1^, only 9% higher, despite a 33% increase in DLI [7]. Based on our regression analysis, we would expect DPIs of 3.2 and 4.3 mol m^−2^ d^−1^ with a 12-h photoperiod and DLIs of 15 and 20 mol m^−2^ d^−1^, respectively. The higher DPIs in our study are at least partly the result of maintaining a stable DLI throughout the photoperiod. Because of the asymptotic relationship between PPFD and ETR, maintaining a constant PPFD results in a higher DPI than variable light levels, assuming the DLI is the same [7]. At a DLI of 17 mol m^−2^ d^−1^, a 12-h photoperiod, and constant PPFD, Weaver and van Iersel [7] estimated the DPI of lettuce to be 3.5 mol m^−2^ d^−1^, consistent with our findings. Weaver and van Iersel [7] also simulated the effect of different photoperiods, all with constant PPFD on DPI. Based on these simulations, using a DLI of 17 mol m^−2^ d^−1^, the DPI is expected to increase from ≈2.8 to 4.4 mol m^−2^ d^−1^, (57%) as the photoperiod increases from 8 to 22 h. We observed substantially greater increases in DPI with longer photoperiods (74 and 109% at DLIs of 15 and 20 mol m^−2^ d^−1^). The greater increase in the current study may be the result of the constant PPFD during the Φ_PSII_ and ETR measurements. Fluctuating PPFD levels require continuous changes in regulation of xanthophyll cycle pigments and the interconversion of violaxanthin and zeaxanthin occurs relatively slowly, over several minutes. Under fluctuating light levels, this can negatively impact electron transport and photosynthesis following a decrease in PPFD [25].

We found a substantially greater increase in DPI with longer photoperiods at a DLI of 20 mol m^−2^ d^−1^, compared a DLI of 15 mol m^−2^ d^−1^. As a matter of fact, at a 7-h photoperiod, the DPI was similar with both DLIs, a result of the response of Φ_PSII_ to PPFD; the 33% higher PPFD (794 vs. 595 μmol m^−2^ s^−1^) with the 7-h photoperiod was offset by a 27% lower Φ_PSII_ (0.28 vs. 0.39 at DLIs of 20 and 15 mol m^−2^ d^−1^, respectively). However, with a 22-h photoperiod, Φ_PSII_ at a DLI of 20 mol m^−2^ d^−1^ (and PPFD of 252 μmol m^−2^ s^−1^) was only 5% lower than that at a DLI of 15 mol m^−2^ d^−1^ (and PPFD of 189 μmol m^−2^ s^−1^), despite the 33% greater PPFD.

### 3.4. Photoperiod and Crop Growth

Longer photoperiods with lower PPFD have been suggested as a strategy to increase photosynthetic efficiency and crop growth in controlled environments [26]. This study confirms that delivering the same DLI over longer photoperiods with lower PPFDs indeed results in greater DPIs in lettuce. Other studies have shown that longer photoperiods with lower PPFDs do translate into more growth. In a growth trial conducted in a growth chamber, Palmer [27] observed an 18% increase in lettuce shoot dry weight as photoperiod increased from 10 to 20 h with a DLI of 16 mol m^−2^ d^−1^. Weaver and van Iersel [8] found that for greenhouse-grown lettuce, dry weight increased by 28% when the photoperiod increased from 12 to 21 h, even though all plants received the same DLI (17 mol m^−2^ d^−1^). In a greenhouse trial with *Rudbeckia fulgida* var. *sullivantii* ‘Goldsturm,’ Elkins and van Iersel [28] determined that shoot and root dry weight increased linearly from 0.23 to 0.30 g/plant (30%) and 0.071 to 0.088 g/plant (24%), respectively, as photoperiods increased from 12 to 21 h, while maintaining a DLI of 12 mol m^−2^ d^−1^.

Much larger increases in DPI (74% to 109%) were observed in this study in response to longer photoperiods, as compared to growth responses (18% to 30%) in other studies. Plants in this study were grown under identical lighting conditions with relatively low PPFD, then transferred for one day to another set of lighting conditions for measurement. Therefore, our plants were not acclimated to the measurement PPFD. If, for example, plants that were measured at a high PPFD and short photoperiod also had been grown at high PPFD, acclimation to high PPFD likely would have resulted in greater photochemical capacity [10], resulting in higher Φ_PSII_, ETR, and DPIs. This would have made treatment differences smaller. Future work should study how acclimation affects DPI. Nonetheless, the combination of our findings with growth chamber [27] and greenhouse studies [8,28] indicates growth rates can be increased with longer photoperiods and lower PPFD.

### 3.5. Increasing Φ_PSII_, ETR, and DPI

Since Φ_PSII_ decreases and ETR increases in response to increasing PPFD, there is an inherent tradeoff between efficient photochemistry and rapid electron transport. This is likely inevitable, because higher PPFD results in a larger fraction of closed PSII reaction centers and upregulation of photoprotective processes. The decrease in Φ_PSII_ with increasing PPFD is largely caused by an upregulation of Φ_NPQ_ (Figure 1). That raises the question of whether this upregulation of photoprotective processes can be reduced. Photoprotective processes are necessary to prevent excessive photoinhibition. Photoinhibition results from excess excitation energy directed towards reaction centers, and most commonly PSII. Excess excitation energy can lead to damage and degradation of the D1 protein in PSII [29]. Although plants can repair this damage, the repair cycle is slow, typically in the order of hours [24]. Photoinhibition results in a decrease of active PSII reaction centers, and thus, reduces the photochemical capacity of leaves. It is therefore not reasonable to suggest that photoprotective processes are not required. However, is it possible to reduce Φ_NPQ_ without inducing excessive photoinhibition?

Plants have evolved under widely fluctuating light conditions, with those fluctuations coming over the course of minutes, hours, days, and seasons. As a result, plants evolved a wide range of photoprotective processes [29]. However, in controlled environmental agriculture it is possible to provide environmental conditions, including light, that are steady and near-optimal for crop growth. Many cultivars grown in controlled environment agriculture were bred for field production, where crops need to yield well under a wide range of environmental conditions; thus, desirable genotypes need to have enough plasticity to achieve phenotypic stability in a range of environments [30]. However, in controlled environment agriculture, the environment can be controlled, and breeding efforts can focus on developing genotypes optimized for that specific environment.

Desirable traits for plants grown in controlled environment agriculture include fast growth, preferably under relatively low PPFD, and high levels of desirable secondary compounds. To achieve fast growth, plants need to form a relatively large canopy quickly so they can intercept the provided light efficiently, and they need to be able to use that provided light efficiently for the production of biomass. Selecting for high Φ_PSII_ seems like a logical target in the breeding of cultivars that can grow well under relatively low light.

Interestingly, the importance of Φ_PSII_ in determining how efficiently plants use light energy to produce biomass is commonly overlooked. Multiple review articles have described the processes involved in the light use efficiency of biomass production in detail, but without addressing the importance of Φ_PSII_ [31,32]. Since energy losses associated with the decrease in Φ_PSII_ in response to increasing PPFD can be large, methods to increase Φ_PSII_ deserve more attention. Such high Φ_PSII_ could potentially be achieved in different ways. One option would be to breed cultivars with a high ratio of photosynthetic reaction centers relative to the size of the light-harvesting antennae. This would increase the overall capacity for electron transport, while reducing the amount of excitation energy directed towards individual reaction centers. Such an approach has been effective in the micro-algae *Dunaliella salina* [33]. This could potentially be coupled with breeding efforts to downregulate light-induced energy dissipation, allowing a larger fraction of the absorbed photons to be used for electron transport. In field-grown tobacco (*Nicotiana tabacum*), transforming plants to respond to fluctuating light levels more quickly by speeding up the conversion between violaxanthin and zeaxanthin and overexpressing PsbS increased CO_2_ assimilation and dry matter production by 15% [25]. This same approach is not likely to benefit plants in plant factories where light levels do not fluctuate, but it indicates that photoprotective mechanisms in crops may not be optimal for maximum growth. Cultivars developed for controlled environments are more likely to benefit from sustained low levels of heat dissipation, coupled with a large capacity for photochemistry. The response of Φ_PSII_ and Φ_NPQ_ are not pre-determined properties of plants; there are large differences among species and plants have the ability to acclimate to their light environment [10]. We have also seen variability among cultivars of the same species (unpublished results); this indicates that there is unexplored genetic variability that can be used as the basis for breeding to increase ΦPSII, ETR, and DPI.

### 3.6. Using DPI to Optimize Lighting Strategies

Lighting recommendations for greenhouse crops are often based on DLI [34,35], regardless of how that DLI is provided. However, crop light use efficiency and growth are affected not only by DLI, but also by how that light is provided over the course of the photoperiod. To account for this, newer lighting control approaches can account for the physiological responses of the crop, such as photosynthesis. For example, the ‘DynaLight Desktop’ control system for high-pressure sodium lights accounts for weather forecasts and variable electricity prices, with the goal of achieving a specific daily photosynthesis integral (similar to DPI, but based on CO_2_ assimilation per unit leaf area, rather than electron transport) [36,37]. This approach resulted in similar crop carbon gain as lighting control based on a PPFD threshold, but with 25% less electricity use. However, under short natural photoperiods, flowering of the long-day crops *Campanula portenschlagiana* ‘Blue Get Mee’ and *Campanula cochlearifolia* ‘Blue Wonder’ was poor and erratic. Thus, control algorithms should also account for photoperiodic flowering responses of crops [38]. The availability of dimmable LED fixtures provides new opportunities for lighting control systems. Taking advantage of the ability of these lights to respond to ambient sunlight conditions in real-time, the lighting strategy can be optimized to achieve a target DPI in a specific photoperiod. When electricity prices do not vary, such a control strategy would result in maintaining a constant PPFD throughout the photoperiod. This was estimated to result in a 7–10% reduction in electricity use, as compared to turning lights on and off based on a PPFD threshold [39]. This approach can be further optimized by including sunlight predictions and the ability to account for variable electricity prices [40].

### 3.7. Conclusions

To our knowledge, this is the first study to quantify DPI over different photoperiods with the same DLI. Lettuce DPI greatly increased with longer photoperiods and lower PPFDs because plants exhibited decreased Φ_PSII_ with increasing PPFD. The short-term physiological responses found in this study are consistent with results from longer-term growth chamber and greenhouse studies where growth was measured [8,28]. This has practical implications for controlled environment agriculture. Longer photoperiods with lower PPFD can increase growth and reduce capital expenses, because fewer light fixtures would need to be installed to provide the appropriate PPFD and DLI. The relationships between PPFD, photoperiod, DLI, DPI, and electricity prices should be taken into account in the development of better lighting control algorithms to assure that lighting is provided to optimize crop growth with the least amount of supplemental light or electricity costs possible. There also appear to be unexplored possibilities to breed cultivars that are better adapted to controlled environment agriculture conditions.

## 4. Materials and Methods

### 4.1. Plant Material

Lettuce (*Lactuca sativa* ‘Green Towers’) seeds were sown in 15-cm diameter round pots filled with a soilless growing medium (Fafard 3B; SunGro Horticulture, Agawam, MA, USA). The seeds were germinated indoors on an ebb-and-flow bench under white light-emitting diode (LED) arrays (Fat Jeff; Aurora, St. Petersburg, FL, USA) with a PPFD of 230 μmol m^−2^ s^−1^ plus 10 μmol m^−2^ s^−1^ of far-red light (700–800 nm). They were fertigated, as needed, with 100 mg L^−1^ N water-soluble fertilizer solution (15N–2.2P–12.5K, Peters Excel 15–5–15 Cal-Mag Special; ICL Fertilizers, Dublin, OH, USA). After 10 d, the plants were thinned to one plant per pot, moved to a growth chamber (E15; Conviron, Winnipeg, Manitoba, Canada), and grown under cool-white fluorescent light with a PPFD of ≈250 μmol m^−2^ s^−1^, a 14-h photoperiod, ambient CO_2_, ≈37% relative humidity, and a constant air temperature of 22.7 °C. Plants were watered using the same fertilizer solution. New seeds were sown every 5 d to maintain a steady supply of plants of similar age. Thirty-six different lettuce plants were used during the study period from 16 January to 6 February 2018.

### 4.2. Experimental Setup

Two DLIs (15 and 20 mol m^−2^ d^−1^) were each applied across six photoperiod treatments (7, 10, 13, 16, 19, and 22 h) to evaluate the effects on Φ_PSII_, ETR, and DPI (Table 1). To measure these parameters, individual plants were moved daily into a second growth chamber (E15, Conviron) that was divided into two separate sections with identical measurement setups. Chlorophyll fluorescence measurements were collected using a modified version of the system described in van Iersel et al. [41]. Each section of the growth chamber was lit using custom-made LED arrays (PhytoSynthetix, Boulder, CO, USA), with the spectral distribution shown in Figure 5. The fixtures were powered by dimmable drivers. The drivers in each section were connected to separate dataloggers (CR1000; Campbell Scientific, Logan, UT, USA). A quantum sensor (LI-190; LI-COR Biosciences, Lincoln, NE, USA) and a chlorophyll fluorometer (MiniPam; Heinz Walz, Effeltrich, Germany) with a leaf clip were placed in each section, and the analog outputs from each were recorded by the datalogger. The datalogger was also connected to the serial communication port of the fluorometer, allowing the datalogger to trigger fluorometer measurements. The PPFDs required to reach the target DLI ranged from 189 to 794 µmol m^−2^ s^−1^, depending on the DLI and photoperiod. To control the PPFD, we programmed the target DLI and photoperiods into the datalogger program before each test. From that information, the datalogger calculated the required PPFD (DLI divided by photoperiod in seconds). Then, based on readings from the quantum sensor, the datalogger sent a voltage signal to the dimmable driver, using an analog output module (SDM-AO4A; Campbell Scientific) to adjust the output from the LED fixtures to achieve the required PPFD. With this setup, we could achieve precise control over the PPFD and DLI; the standard deviation of the mean PPFD for all 36 tests averaged 0.1 µmol m^−2^ s^−1^, while the DLI in each run was within 0.001 mol m^−2^ d^−1^ of the target DLI.

Two mature plants, approximately four weeks old, were selected each day data were collected. One plant was placed in each section and a fully expanded, uppermost leaf was chosen for chlorophyll fluorescence measurements. The leaf clip was attached to the leaf and the plant was kept in the dark for a minimum of 30 min to adapt to the dark, after which the dark-adapted steady-state (F_o_) and maximum fluorescence (F_m_) were measured. After the lights came on, steady-state (F_t_) and maximum (F_m_^’^) fluorescence were measured every 15 min using the approach described by van Iersel et al. [41] throughout the entire photoperiod. Immediately after F_m_^’^ was measured, the actinic light was turned off and far-red LEDs (peak at 735 nm) were turned on to oxidize the plastoquinone pool and open all PSII reaction centers, after which F_0′_ was measured. Using F_m_, F_m_^’^, and F_t_, Φ_PSII_, NPQ, and ETR were calculated [4]. The ETR was calculated assuming a leaf absorptance of 0.84 [12]. Both Φ_NO_ and Φ_NPQ_ were calculated assuming a lake model organization of light harvesting antennae and reaction centers [21]. During the measurement period, air temperature was 22.8 ± 0.2 °C (mean ± sd), vapor pressure deficit was 1.7 ± 0.2 kPa, and CO_2_ level was ambient inside the growth chamber.

### 4.3. Experimental Design and Data Analysis

The six photoperiods were randomized between days. Once the photoperiod for a specific day was determined, the DLIs of 15 and 20 mol m^−2^ d^−1^ were randomly assigned to the two sections of the growth chamber. Each treatment combination (DLI × photoperiod) was repeated three times. SigmaPlot (version 11.0; Systat Software, San Jose, CA) was used to analyze all the data. The effect of PPFD on Φ_PSII_ and ETR were tested using linear and asymptotic rise to a maximum regression analyses [7]. To account for potential effects of DLI on the relationship between PPFD and Φ_PSII_ or ETR, the DLI was included in the initial regression model as well, but was not significant.

The effect of photoperiod and DLI on DPI was tested using multiple regression, with photoperiod, DLI, and their interaction included in the initial model. Non-significant effects (main effects of photoperiod and DLI) were then removed using backward selection (α = 0.05).

## Figures and Tables

**Figure 1 plants-09-01172-f001:**
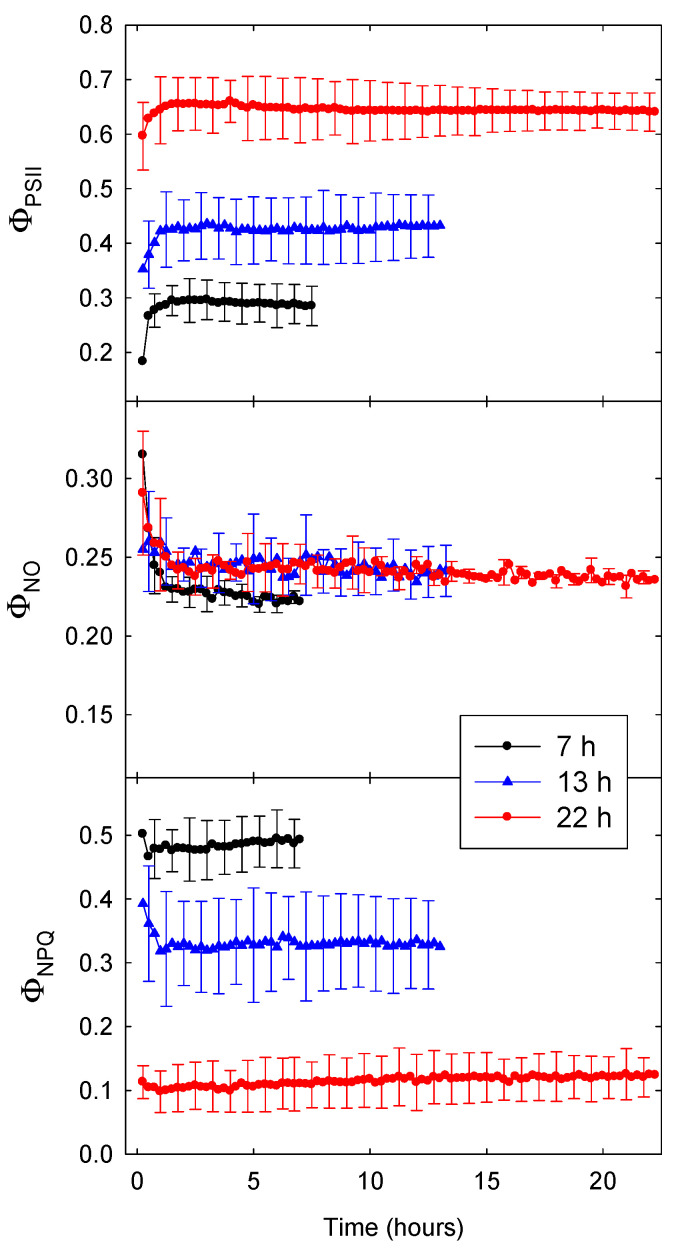
The quantum yield of photosystem II (Φ_PSII_, top), the quantum yield of non-light-induced energy dissipation (Φ_NO_, middle), and the quantum yield of light-induced energy dissipation (Φ_NPQ_, bottom) of lettuce (*Lactuca sativa* ‘Green Towers’) under three different photoperiods (7, 13, and 22 h) with a daily light integral (DLI) of 20 mol m^−2^ d^−1^. Data are the means of three replications ± SD.

**Figure 2 plants-09-01172-f002:**
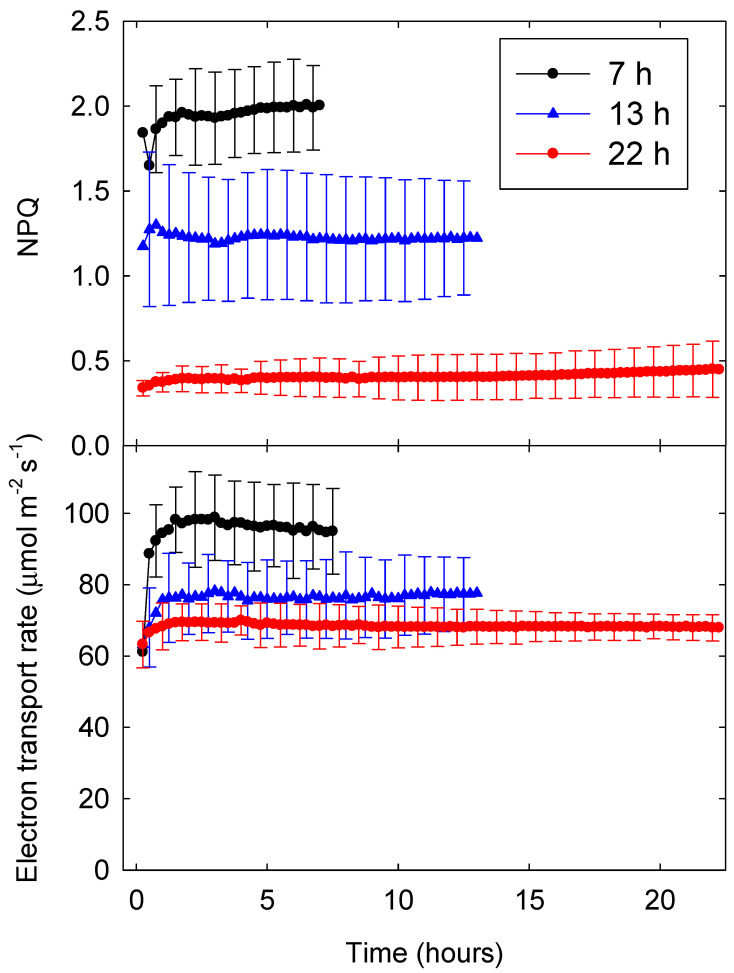
Non-photochemical quenching (NPQ, top) and electron transport rate (ETR, bottom) of lettuce (*Lactuca sativa* ‘Green Towers’) under three different photoperiods (7, 13, and 22 h) with a daily light integral (DLI) of 20 mol m^−2^ d^−1^ over the course of the photoperiod. The area under each ETR curve represents the daily photochemical integral (DPI). Data are the means of three replications ± SD.

**Figure 3 plants-09-01172-f003:**
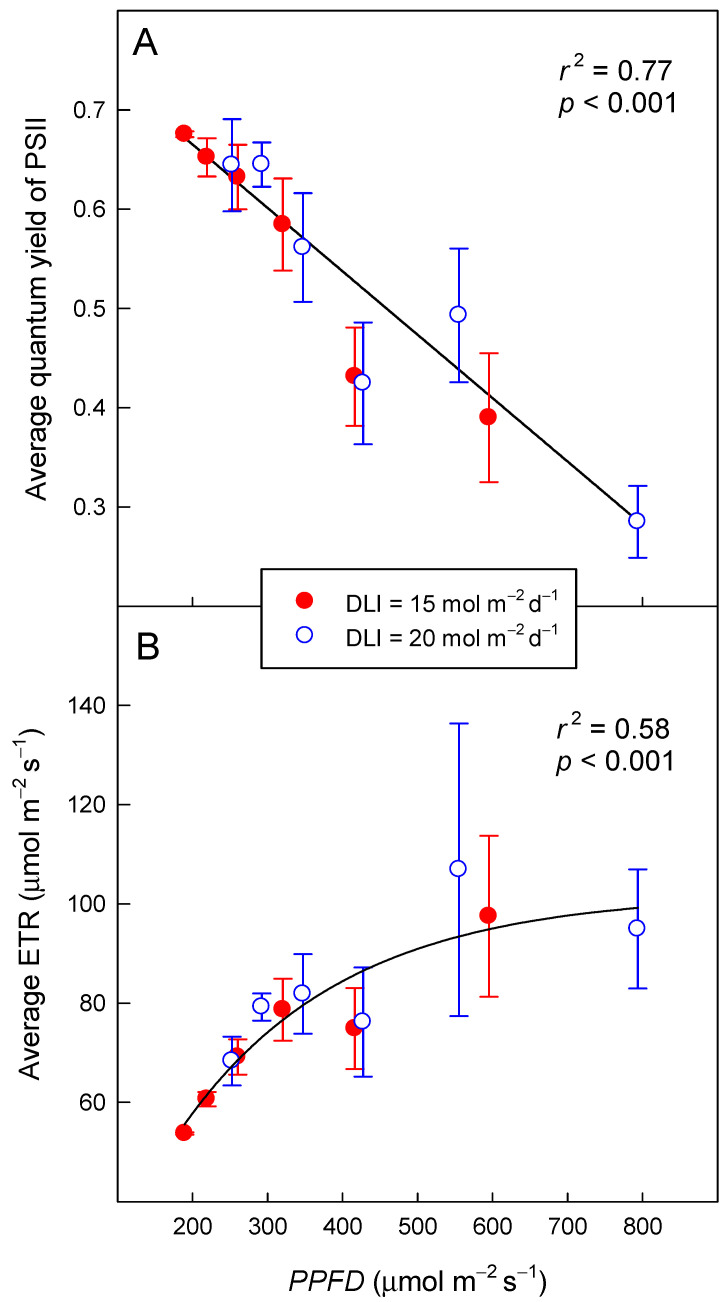
The quantum yield of photosystem II (Φ_PSII_ = 0.792 − 0.000663 × PPFD) (**A**) and the electron transport rate [ETR = −10.1 + 112.1 × (1 − e^−0.00463 × PPFD^)] (**B**) of lettuce (*Lactuca sativa* ‘Green Towers’), averaged over the entire photoperiod, as a function of photosynthetic photon flux density (PPFD) Data points represent the mean ± SD (n = 3). Plants were measured under two daily light integrals (DLI), 15 and 20 mol m^−2^ d^−1^, but DLI had no effect on the relationship between PPFD and these parameters.

**Figure 4 plants-09-01172-f004:**
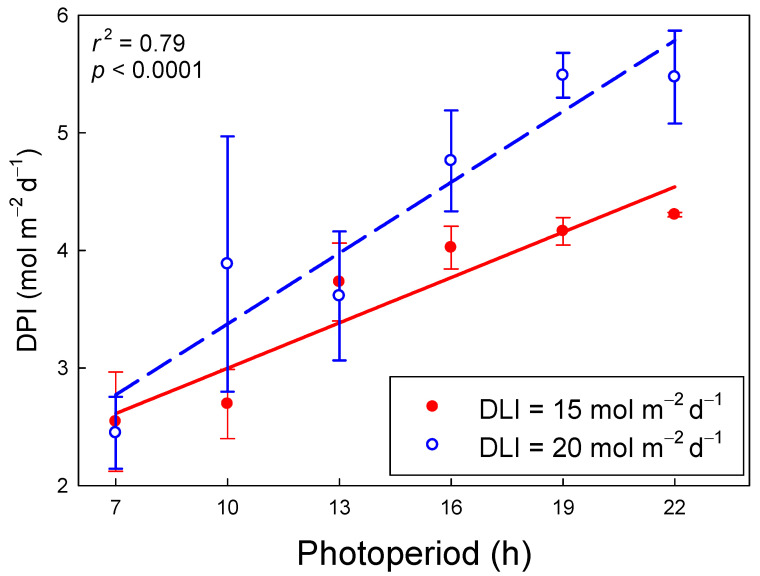
The daily photochemical integral (DPI) of lettuce (*Lactuca sativa* ‘Green Towers’) as a function of photoperiod and daily light integral (DLI) [DPI = 1.508 + (0.00953 × photoperiod × DLI)]. Plants measured under longer photoperiods, or under the same photoperiod but at a lower DLI, received a lower photosynthetic photon flux density (PPFD) (Table 1). There was a significant interaction (*p* < 0.001) between DLI and photoperiod as indicated by the different slopes of the regression lines.

**Figure 5 plants-09-01172-f005:**
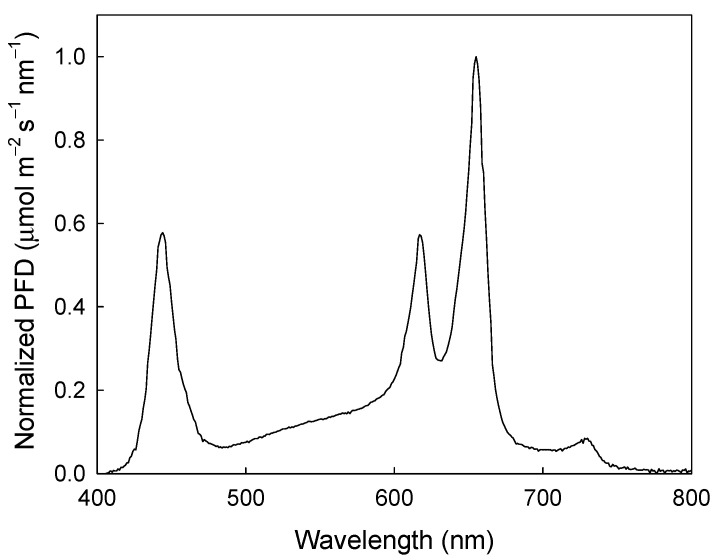
Normalized photon flux density (PFD) of the custom-made light-emitting diode (LED) arrays used for photochemical measurements.

**Table 1 plants-09-01172-t001:** Photosynthetic photon flux density (PPFD) during photochemical measurements. Plants were measured during photoperiods ranging from 7 to 22 h and PPFD was adjusted to ensure a daily light integral (DLI) of 15 or 20 mol m^−2^ d^−1^.

Photoperiod (h)	15 mol m^−2^ d^−1^	20 mol m^−2^ d^−1^
	—PPFD (µmol m^−2^ s^−1^)—	
7	595	794
10	416	555
13	320	427
16	260	347
19	219	292
22	189	252

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
