# Peer review of "Longer Photoperiods with the Same Daily Light Integral Increase Daily Electron Transport through Photosystem II in Lettuce"

_plants, 2020, doi:10.3390/plants9091172_

Round 1

Reviewer 1 Report

Using DPI as a metric to assess light use and comparing this to DLI is novel and will significantly add to the toolbox to assess optimum lighting for crops. A few comments:

Figure 1. It is really interesting how low the PSII QY and how much higher the NPQ QY are at high and moderate PPFD with shorter PPs. Also, the plants are acclimating only slightly as seen by the transients early on.

Figure 2. NPQ follows the same trend as the NPQ QY. The SDs for electron transport rates are really high and overlap considerable. Are these difference ET rates statistically significant?

Line 122. Should this read…0.663 umol m-2 s-1 rather than mmol?

Figure 3. I did not see where the equations came from for  fPSII and ETR in the methods. Can you please clarify and cite source?

Figure 4. I had the hardest time with this figure. Can the equation for DPI be described in the methods section? I have consulted a couple of data analysts as these data do not seem to fit a linear regression. They look more sigmoidal, especially for the DLI of 15. The DPI for PP10 at a DLI of 20 is obviously an outlier as the SD is so high. I am not sure what order this curve fit would be off hand but it should be modeled differently. Can this be remodeled to better reflect the data?

Discussion:

I found the discussion too long. I think the section from line 165 to line 178 can be removed  at the sentence ending in …stabilization of photochemistry.

Lines 178-182. I do not see an upregulation of NPQ QY (Fig 1), it actually decreased within the first hour. NPQ only had slight upregulation for 13h PP (Fig 2). The text should reflect this accordingly.

Line 202-204. Shade grown plants have larger LHCs therefore there would be a greater decrease in fPSII at higher light when plants were shifted as in these experiments.  

Line 233-234. I don’t understand why an increase in DPI with increasing PP and DLI was expected due to a decrease in fPSII. Can you please explain?

Line 244. DLI should be replaced with DPI.

Line 250-253. Omit sentence about fluctuating light. This experiment is not looking at fluctuating light but a shift in PPFD and PP?

Line 311- 318. There are quite a few papers that describe crop selection based on increased phoitosynthetic capacity. Can these be cited here? Don Ort and Stephen Long have published a few. Also, Anastasios Melis has published about increasing photosynthesis and reducing photodamage based on truncating the LHCs.

Line 320. Do you mean zeaxanthin rather than antheraxanthin here?

Line 327-329. See above, there are papers published on this.

Section 3.6. There is no real discussion about DPI. I would like to see more here.

In general, there was a lot of repetition about photoinhibition and damage throughout the discussion. Can this be summed up once where it would be most relevant?

Conclusions. Is there any data on DPI and yield?

Methods: Line 406-407. I think you mean the PQ pool is oxidized rather than reduced?

There should be descriptions here regarding the equations that were used throughout.

References: Ref 29 and 31 are repeats. Please remove one of them.

Reviewer 2 Report

The article describes an experiment that demonstrates that the total electron transport through photosystem II integrated over 24 hours increases as the photoperiod is increased after holding DLI constant.

The article is very well written, and I enjoyed reading it: the methodology is solid, the results are well described, the literature review is sufficient...  The remaining question is whether or not its contribution to the existing literature is sufficient enough to justify publication in this journal. 

In particular, it is already understood that quantum yield of PSII decreases with PPFD. For example, this article discusses how Weaver and van Iersel (2019) show that quantum yield decreases and ETR increases at a decreasing rate as PPFD increases. So, in the case where DLI is constant and we have three PPFD treatments, if we estimate quantum yield every 15 minutes then integrate that curve, is there any question that the area under that curve would decrease with PPFD?  Under which circumstances would the results in this article contradict those presented in Weaver and van Iersel (2019)? 

When I started reading the manuscript my expectation was that the results would give insights into how much these differences in DPI could matter to a grower.  Weaver and van Iersel (2019) gave a sense of the opportunity offered by extending photoperiod to increase photosynthetic efficiency using dry weight differences.  Does an understanding of how PPFD impacts the intraday dynamics of ETR, NPQ, and quantum yield of PSII offer a new opportunity for increasing the profitability of growing using artificial lighting?

I suspect this comes down to a few sentences that better clarify the article’s differentiation from previous research and how the estimated dynamics improve our capacity to reduce energy costs or increase yields.
